# Falls and long-term survival among older adults residing in care homes

**Alicia Padrón-Monedero**[1,2,3], **Roberto Pastor-Barriuso**[1,4], **Fernando J. García López**[1,3], **Pablo Martínez Martín**[1,3], **Javier Damián**[1,3]*

**1** National Centre for Epidemiology, Carlos III Institute of Health, Madrid, Spain, **2** Department of Preventive Medicine and Public Health, School of Medicine, Universidad Autónoma de Madrid/ IdiPAZ, Madrid, Spain, **3** Consortium for Biomedical Research in Neurodegenerative Diseases (*Centro de Investigación Biomédica en Red sobre Enfermedades Neurodegenerativas—CIBERNED*), Madrid, Spain, **4** Consortium for Biomedical Research in Epidemiology and Public Health (*Centro de Investigación Biomédica en Red en Epidemiología y Salud Pública—CIBERESP*), Madrid, Spain

* jdamian@isciii.es

## Abstract

### Objectives

To assess the association between having suffered a fall in the month prior to interview and long-term overall survival in nursing-home residents.

### Methods

Retrospective cohort study conducting an overall survival follow-up of 689 representative nursing-home residents from Madrid, Spain. Residents lived in three types of facilities: public, subsidized and private and its information was collected by interviewing the residents, caregivers and/or facility physicians. Residents contributed to follow-up time from their baseline interviews until death or being censored at the end of the 5-year follow-up period. The association between suffering a fall during the month prior to interview and long-term overall survival was analyzed using Cox proportional hazards models. To adjust for potential confounders we used progressive adjusted models. We then repeated the analyses with severity of the fall (no fall, non-severe, severe) as the main independent variable.

### Results

After a 2408 person-year follow-up (median 4.5 years), 372 participants had died. In fully-adjusted models, residents who had suffered any kind of fall in the previous month showed virtually the same survival rates compared to non-fallers (hazard ratio (HR) = 1.03; 95% CI = 0.75–1.40). There was a weak graded relationship between increased fall severity and survival rates for the non-severe fall group (HR = 0.92; 95% CI = 0.58–1.45) and the severe fall group (HR = 1.36; 95% CI = 0.73–2.53) compared with residents who had not suffered any kind of fall. The hazard ratios for severe falls were higher in men, residents with less comorbidity, fewer medications, and those functionally independent.

**Data Availability Statement:** The data for our project have been deposited to Institutional repository of the Carlos III Institute of Health,

Repisalud at http://hdl.handle.net/20.500.12105/9416.

**Funding:** This work was supported by the Carlos III Institute of Health (grant number, PI15CIII00037) to JD. The funders had no role in study design, data collection and analysis, decision to publish, or preparation of the manuscript.

**Competing interests:** The authors have declared that no competing interests exist.

## Conclusion

We found no associations between having suffered a fall in the month prior to interview and long-term survival; neither did we find a marked association when severity of fall was accounted for in the whole population. In some subgroups, however, the results merit further scrutiny.

## Introduction

Older people's accidental falls are a priority for WHO/Europe [1] and they constitute a major public health problem. In the European Union, older adults account for half of deaths from unintentional injuries [2], despite representing one fifth of the population [3]. In older adults from developed countries, unintentional injuries are between the fifth and the seventh most common causes of death [4, 5], and the majority of unintentional injuries are caused by falls [2, 4, 6]. Furthermore, mortality due to accidental falls has recently increased in older adults [7–10], and the rapid ageing of the population makes it probable that this trend will increase in the future [7, 11]. Older adults have significantly higher mortality from minor injuries than those under 50s [12] and the interval from injury to death could be longer for minor or moderate injuries (for example rib fracture and chest or muscle contusions) than for more severe ones (hip, vertebral or skull fractures) [13]. Additionally, as the interval between injury and death increases, fall mortality could be underreported due to the death being attributed to a concomitant pathology [13, 14] (pneumonia, cerebral hemorrhage, pulmonary thromboembolism, worsening of comorbidities or other complication of the fall).

Despite the substantial number and significant consequences of aging adults' accidental falls, it is remarkable that a relatively limited number of recent studies have analyzed the long-term mortality of accidental falls in older adults. Moreover, most of these studies come from hospital settings [15–18] rather than community settings [19] and the scant literature that compares the long-term mortality risk after a fall with an appropriate control group is either not recent [20–23] or limited to women [24]. The study of long-term survival could add further insight into the real burden of this important problem, particularly in nursing homes where falls are frequent; as far as we know, no recent study has properly focused on falls' long-term consequences in a representative sample of nursing home residents. Hence, the aim of this study is to analyze the association between the baseline presence of a fall, and the severity of the fall, with long-term overall survival of care home residents.

## Materials and methods

### Study population

This cohort study used baseline data from a survey conducted from June 1998 through June 1999 in a representative sample of residents aged 65 years or older in residential and nursing homes in the city of Madrid, Spain, and a surrounding area of 35 km. Study participants were selected through stratified cluster sampling, one stratum included 47 public or subsidized (privately owned but publicly funded) nursing homes and the other stratum included 139 private institutions. We initially selected 25 public/subsidized and 30 private institutions with probability of selection proportional to their size (range 16 to 620 beds), and then sampled 10 men and 10 women from each public/subsidized facility selected and 5 men and 5 women from each private facility. The subjects were selected by systematic sampling with random start,

using facilities' complete alphabetical lists of residents. Four private institutions declined to participate (totaling 40 sample subjects) and 45 additional residents could not be selected due to prolonged absence or refusal, leading to an overall response rate of 89% (715 out of 800 sample residents). Thirty-nine subjects were randomly substituted with residents of the same facility and sex, yielding a total of 754 residents. As a result of this design, residents in public/subsidized facilities and men were oversampled. To correct for this oversampling, sampling weights were assigned to study participants as the inverse of their probabilities of selection (that is, the inverse of the sampling fractions by type of facility and sex) [25], so that the weighted distribution of these factors in the study sample matched their distribution in the entire population of care home residents.

The Research Ethics Committee of the Institute of Health Carlos III approved the study. Informed consent was obtained verbally from all participants or their closest relative and documented.

## Baseline data collection

Trained geriatricians or residents in geriatrics administered structured questionnaires verbally to all selected residents, their main caregivers, and facility physicians to collect baseline data on sociodemographic characteristics, medical conditions, prescribed medications, and functional dependency. Facility physicians (or nurses for 8% of residents) were asked about the occurrence of any fall during the previous month. Where the answer was one or more, they were then asked to state whether any of the falls had any of the following consequences: open wound, hip fracture, other fracture, cerebral hemorrhage, or transfer to hospital. Falls with any of these consequences were classified as severe; otherwise, as non-severe [26].

Chronic medical conditions–including cancer, chronic obstructive pulmonary disease, arrhythmias, ischemic heart disease, congestive heart failure, peripheral arterial disease, stroke, hypertension, diabetes, anemia, Alzheimer's disease, other dementias, Parkinson's disease, epilepsy, depression, anxiety disorders, and arthritis–were ascertained by interviewing facility physicians with access to medical histories. Dementia was defined as a physician's diagnosis of Alzheimer's disease or other dementias, and the number of chronic conditions other than dementia was computed. Medications used in the preceding 7 days were obtained by reviewing medical records. Use of antidepressants (code N06A of the World Health Organization Anatomical Therapeutic Chemical Classification) was recorded, and the number of prescribed medications other than antidepressants was also noted.

Urinary (in)continence was assessed by asking the study subjects or their main caregivers (if assigned, 41%) if the resident had experienced any involuntary leakage of urine in the preceding 14 days, and classified into no, mild (only nocturnal or occasional diurnal leakage), or severe urinary incontinence (frequent or total diurnal leakage) [27]. Functional dependency when performing basic daily living activities was evaluated by using a modified Barthel index [28]. For this study, we used a version that excluded incontinence and which has been used in previous studies. The index's score ranged from 0 to 80 points [29]. Residents were classified as: functionally independent (80 points), mildly/moderately dependent (31–79 points), or severely/totally dependent (0–30 points).

## Mortality follow-up

We followed up study participants for mortality through to 15 September 2013, but only the first 5 years of follow-up were used in the present study. Our rationale for a 5-year follow-up was choosing a timeframe after which the occurrence of an event was hardly attributable to any change at baseline. All-cause mortality was ascertained by requesting updated data on

residents' vital status from participating facilities and through computerized linkage to the Spanish National Death Index, which includes all deaths registered in Spain [30]. Residents contributed follow-up time from their 1998–1999 baseline interviews until death or being censored at the end of the 5-year follow-up period.

## Statistical analysis

Hazard ratios (HR) for all-cause mortality and 95% confidence intervals (CIs) by occurrence (yes vs. no) and severity of fall (non-severe or severe vs. no) were estimated using Cox proportional hazards models with two increasing levels of adjustment. The first model included baseline age (65–74, 75–79, 80–84, 85–89, or $\geq$ 90), sex (woman or man), and type of facility (public, subsidized, or private), and the second model further adjusted for baseline dementia (yes or no), number of chronic conditions other than dementia (0–1, 2–3, or $\geq$ 4), use of antidepressants (yes or no), number of prescribed medications other than antidepressants (0–2, 3–4, or $\geq$ 5), urinary incontinence (no, mild, or severe), and functional dependency without incontinence (independent, mild/moderate, or severe/total). To explore potential variations in the association between severity of fall and all-cause mortality by resident subgroups, we included interactions between fall-severity categories and subgroups in fully-adjusted Cox proportional hazards models.

Adjusted non-parametric cumulative mortality curves for residents with no, non-severe, and severe falls were estimated as the complement of the baseline survival functions from a Cox model stratified by severity of fall and adjusted to the overall weighted percentages of all baseline confounders. To allow for time-dependent effects of fall severity on mortality (that is, non-proportional hazards), smooth cumulative hazard ratios over time and 95% CIs were estimated from a spline-based parametric survival model [31], with the baseline log cumulative hazard as a natural cubic spline of log time with a single internal knot at the median failure time, interactions of spline terms with fall-severity categories, and adjustment for all baseline confounders. Proportional hazards were contrasted by using joint Wald tests for spline-by-severity interaction coefficients.

Due to the complex sampling design and the different selection probabilities of study participants, all analyses were weighted by the inverse of each participant's selection probability to obtain unbiased estimates of population parameters and accounted for the effect of stratification and clustering on standard errors. All reported $P$ values were two-sided. Analyses were carried out using Stata, version 14.2 (StataCorp LP, College Station, Texas 77845 USA) and R, version 3.4 (R Foundation for Statistical Computing, Vienna, Austria).

## Results

Of the 754 participants in the baseline survey, we excluded 10 residents (1.3%) with missing information regarding falls at baseline and 55 residents (7.3%) with unknown vital status at the end of follow-up, thus leaving a final cohort of 689 residents. There were 12.6% of residents who reported at least one fall in the 30 days preceding interview, and 29.5% of those who fell suffered adverse outcomes; the severe fall group was comprised of these subjects. Fallers presented more comorbidities and prescribed medications, had higher prevalences of antidepressant use and urinary incontinence, and tended to have a higher degree of functional dependency than residents who had not fallen (Table 1).

After a median follow-up of 4.5 years, 53 fallers had died during 278 person-years of follow-up (mortality rate 18.0 per 100 person-years) and 319 non-fallers died during 2,130 person-years (mortality rate 14.7 per 100 person-years). In a model which was fully adjusted for socio-demographic and clinical characteristics, the risk for all-cause mortality was similar among

**Table 1. Baseline characteristics of nursing home residents by severity of fall (1998–1999)[a].**

| | | | Fall | | |
|---|---|---|---|---|---|
| Characteristic | Overall | No | Non-severe | Severe | P value[b] |
| No. of residents | 689 (100) | 602 (87.4) | 60 (8.9) | 27 (3.7) | |
| Age (years) | | | | | 0.60 |
| 65–74 | 91 (12.2) | 80 (11.9) | 9 (16.7) | 2 (8.6) | |
| 75–79 | 109 (14.7) | 96 (14.7) | 8 (11.8) | 5 (20.5) | |
| 80–84 | 176 (25.5) | 159 (26.6) | 12 (16.7) | 5 (21.8) | |
| 85–89 | 174 (26.8) | 146 (25.9) | 21 (37.3) | 7 (23.2) | |
| ≥ 90 | 139 (20.7) | 121 (20.8) | 10 (17.5) | 8 (25.9) | |
| Sex | | | | | 0.83 |
| Women | 379 (75.6) | 330 (75.6) | 35 (77.2) | 14 (71.9) | |
| Men | 310 (24.4) | 272 (24.4) | 25 (22.8) | 13 (28.1) | |
| Type of facility | | | | | 0.48 |
| Public | 392 (46.4) | 347 (46.5) | 31 (46.8) | 14 (43.8) | |
| Subsidized | 72 (8.1) | 58 (7.6) | 9 (9.0) | 5 (17.0) | |
| Private | 225 (45.5) | 197 (45.8) | 20 (44.2) | 8 (39.2) | |
| Dementia | | | | | 0.67 |
| No | 488 (68.3) | 430 (69.0) | 41 (64.2) | 17 (63.9) | |
| Yes | 197 (31.7) | 168 (31.0) | 19 (35.8) | 10 (36.1) | |
| Unknown | 4 | 4 | 0 | 0 | |
| No. of chronic conditions[c] | | | | | < 0.001 |
| 0–1 | 180 (26.7) | 171 (29.4) | 7 (9.4) | 2 (4.8) | |
| 2–3 | 289 (41.9) | 256 (42.7) | 23 (37.6) | 10 (35.7) | |
| ≥ 4 | 220 (31.3) | 175 (27.9) | 30 (53.0) | 15 (59.5) | |
| Use of antidepressants | | | | | < 0.001 |
| No | 607 (88.9) | 540 (91.1) | 43 (67.7) | 24 (88.0) | |
| Yes | 67 (11.1) | 48 (8.9) | 16 (32.3) | 3 (12.0) | |
| Unknown | 15 | 14 | 1 | 0 | |
| No. of prescribed medications[d] | | | | | < 0.001 |
| 0–2 | 174 (26.5) | 167 (29.4) | 4 (5.8) | 3 (8.0) | |
| 3–4 | 238 (36.3) | 214 (37.4) | 16 (23.8) | 8 (41.3) | |
| ≥ 5 | 250 (37.2) | 196 (33.2) | 38 (70.4) | 16 (50.7) | |
| Unknown | 27 | 25 | 2 | 0 | |
| Urinary incontinence | | | | | 0.01 |
| No | 328 (46.2) | 305 (49.2) | 16 (22.2) | 7 (31.5) | |
| Mild | 121 (18.9) | 98 (17.5) | 16 (31.7) | 7 (22.2) | |
| Severe | 218 (34.9) | 183 (33.4) | 24 (46.1) | 11 (46.3) | |
| Unknown | 22 | 16 | 4 | 2 | |
| Functional dependency[e] | | | | | 0.06 |
| Independent | 224 (28.1) | 209 (30.2) | 12 (15.8) | 3 (8.3) | |
| Mild/moderate | 309 (49.3) | 259 (47.7) | 34 (58.6) | 16 (64.1) | |
| Severe/total | 138 (22.6) | 118 (22.0) | 13 (25.6) | 7 (27.6) | |
| Unknown | 18 | 16 | 1 | 1 | |

[a] Unweighted sample counts (weighted percentages).

[b] P value for homogeneity of weighted percentages across categories of severity of fall.

[c] Number of chronic conditions other than Alzheimer's disease and other dementias.

[d] Number of prescribed medications other than antidepressants.

[e] Functional dependency excluding continence criteria.

**Table 2. Hazard ratios for all-cause mortality by occurrence and severity of fall among nursing home residents in Madrid, Spain.**

| Fall | No. of deaths | No. of person-years | Mortality rate[a] (95% CI) | Hazard ratio[b] (95% CI) | |
|---|---|---|---|---|---|
| | | | | Model 1[c] | Model 2[d] |
| No | 319 | 2129.5 | 14.7 (13.1–16.6) | 1.00 (reference) | 1.00 (reference) |
| Yes | 53 | 277.7 | 18.0 (13.4–24.4) | 1.20 (0.90–1.62) | 1.03 (0.75–1.40) |
| Non-severe | 34 | 204.6 | 16.4 (11.5–23.5) | 1.11 (0.71–1.72) | 0.92 (0.58–1.45) |
| Severe | 19 | 73.1 | 22.8 (12.8–40.5) | 1.49 (0.77–2.88) | 1.36 (0.73–2.53) |
| *P* for trend[e] | | | | 0.14 | 0.50 |

[a] Weighted mortality rates per 100 person-years and 95% confidence intervals (CIs).

[b] Hazard ratios for all-cause mortality and 95% confidence intervals (CIs) were obtained from Cox proportional hazards models with follow-up restricted to 5 years and accounting for the stratified cluster sampling and the different selection probabilities.

[c] Model 1 was adjusted for baseline age (65–74, 75–79, 80–84, 85–89, or ≥ 90 years), sex (women or men), and type of facility (public, subsidized, or private).

[d] Model 2 was further adjusted for baseline dementia (yes or no), number of chronic conditions other than dementia (0–1, 2–3, or ≥ 4), use of antidepressants (yes or no), number of prescribed medications other than antidepressants (0–2, 3–4, or ≥ 5), urinary incontinence (no, mild, or severe), and functional dependency excluding (in)continence (independent, mild/moderate, or severe/total).

[e] *P* value for log-linear trend in hazard ratios across categories of severity of fall.

fallers and non-fallers (HR = 1.03; 95% CI = 0.75–1.40) (Table 2). When falls were broken down by their severity, the hazard ratios (95% CIs) for all-cause mortality were 0.92 (0.58–1.45) for residents with non-severe falls and 1.36 (0.73–2.53) for residents with severe falls, both as compared to those without reported falls. This moderate increase in risk among severe fallers was also observed in the adjusted cumulative mortality curves by severity of fall, although the reduced number of residents with severe falls resulted in a somewhat jagged and imprecise curve (Fig 1).

Time-dependent effects of falls on mortality are shown in Fig 2. The cumulative hazard ratio comparing non-severe falls with no fall remained close to the null effect over the entire follow-up (*P* for proportional hazards = 0.94). However, residents with severe falls appeared to experience an increased risk of death during the first 2 years of follow-up, reaching a 2-year cumulative hazard ratio of 1.54 (95% CI = 0.86–2.75), with a progressive decrease in risk thereafter, albeit this time-dependent pattern was not significant (*P* for proportional hazards = 0.70).

In subgroup analyses, non-severe falls were associated with a 69% increase in mortality risk for residents in private facilities and a 37% decrease in risk for those in public institutions (Fig 3). The hazard ratios for severe falls were higher in men (2.00), residents with less comorbidities (1.87), fewer prescribed medications (2.01), and those who were functionally independent (2.10).

## Discussion

We found no notable association between reporting a fall of any type in the month prior to baseline and long-term overall survival among older adults residing in care homes. However, there was a small non-significant increase in risk among severe fallers. They appeared to experience a moderately increased risk of death during the first 2 years of follow-up, with a progressive decrease in risk thereafter. Furthermore, men, residents with fewer comorbidities, fewer prescribed medications and those who were functionally independent; who had suffered a severe fall, did have a noticeably increased risk of mortality compared to residents with no fall in the month prior to the baseline interview.

A limited number of studies from non-clinical settings have assessed long-term overall survival in older adults who had suffered a fall. Moreover, few of these studies compared survival

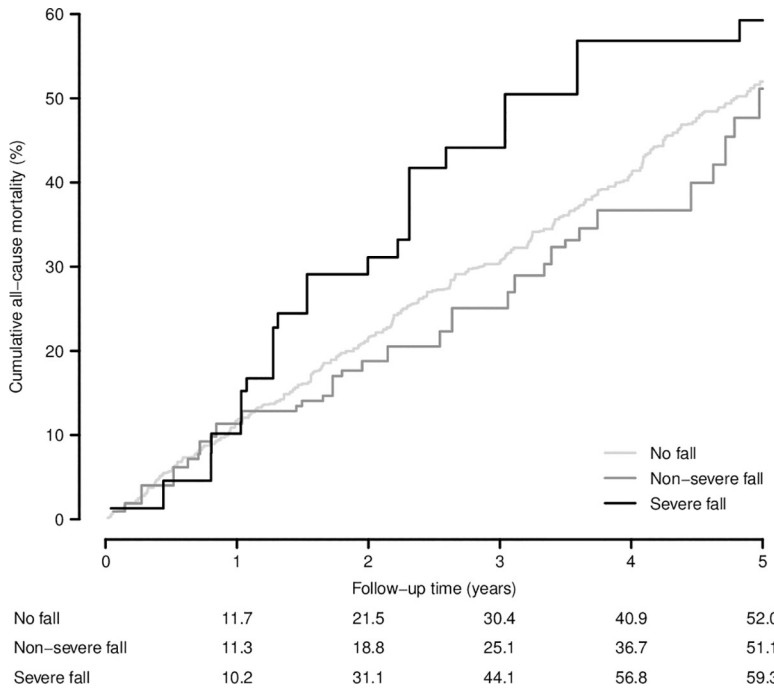

| | | | | | |
|---|---|---|---|---|---|
| No fall | 11.7 | 21.5 | 30.4 | 40.9 | 52.0 |
| Non–severe fall | 11.3 | 18.8 | 25.1 | 36.7 | 51.1 |
| Severe fall | 10.2 | 31.1 | 44.1 | 56.8 | 59.3 |

**Fig 1. Adjusted cumulative all-cause mortality by severity of fall among nursing home residents in Madrid, Spain.**
Non-parametric cumulative mortality curves were obtained as the complement of the baseline survival functions from a Cox model stratified by severity of fall (no, non-severe, or severe), adjusted to the overall weighted percentages of baseline age (65–74, 75–79, 80–84, 85–89, or ≥ 90 years), sex (women or men), type of facility (public, subsidized, or private), dementia (yes or no), number of chronic conditions other than dementia (0–1, 2–3, or ≥ 4), use of antidepressants (yes or no), number of prescribed medications other than antidepressants (0–2, 3–4, or ≥ 5), urinary incontinence (no, mild, or severe) and functional dependency excluding (in)continence (independent, mild/moderate, or severe/total), and accounting for the stratified cluster sampling and the different selection probabilities.

against an appropriate control group of non-fallers, and those that did were not published recently or were limited to women. Sylliaas *et al.*'s studied women aged 75 or over living in the community [24]. They found that women who had suffered at least two falls had poorer survival rates (HR = 1.6; 95% CI = 1.1–2.4) compared to non-fallers after a 9 year follow-up. However, no associations were found between suffering only one fall and subsequent survival (HR = 1.1; 95% CI = 0.7–1.7) [24]. Donald *et al.* found, that all-cause mortality did not increase for single fallers compared to non-fallers (HR = 0.97; 95% CI = 0.7–1.4), in a community cohort of adults over 75 [20]. This is consistent with our results, although we did not assess the impact on survival of subsequent falls during the follow-up period. Altuough, Donald *et al.* did find associations between falling more than once during the 3-year follow-up and mortality (HR = 1.9; 95% CI = 1.2–3.0).[20] Along the same lines, Bath *et al.*'s community-based study found associations between frequent fallers (≥3) and mortality (4-year follow-up) [22]. Nevertheless, they found no significant increase in mortality for older adults who had fallen once or twice compared to non-fallers [22]. In an earlier study from 1992, Dunn *et al.* studied a community sample of adults ≥70 years with a 2 year follow-up [21]. They did not find any associations between suffering either one or multiple falls and mortality, after adjusting for the main covariates (including disability and chronic conditions). The HR were 1.3 (95% CI = 0.9–1.7) for a single fall and 1.3 (95% CI = 0.9–1.8) for multiple falls, as compared to an appropriate control group of non-fallers [21].

Bailly *et al.* studied a cohort of 329 community-based women who had suffered a fall with a 13 year follow-up. They found that elderly women who had suffered an indoor fall had shorter

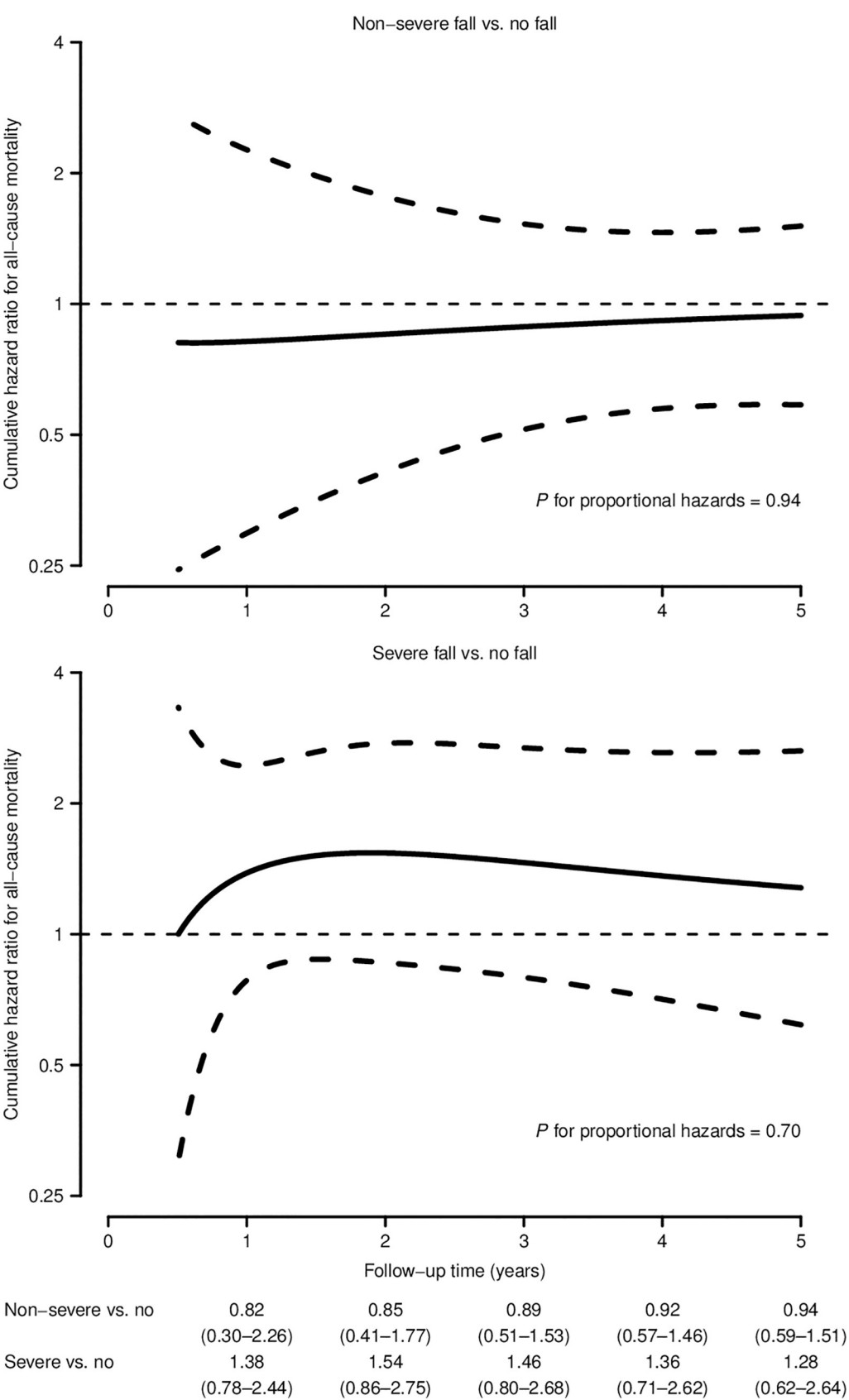

**Fig 2. Cumulative hazard ratios for all-cause mortality over time by severity of fall among nursing home residents in Madrid, Spain.** Smooth cumulative hazard ratios over time (solid curves) and 95% confidence intervals (dashed curves) were obtained from a spline-based parametric survival model with a single internal knot at the median failure time, stratified by severity of fall (no fall, non-severe, or severe), adjusted for baseline age (65–74, 75–79, 80–84, 85–89, or ≥ 90 years), sex (women or men), type of facility (public, subsidized, or private), dementia (yes or no), number of chronic conditions other than dementia (0–1, 2–3, or ≥ 4), use of antidepressants (yes or no), number of prescribed medications other than antidepressants (0–2, 3–4, or ≥ 5), urinary incontinence (no, mild, or severe) and functional dependency without (in)continence (independent, mild/moderate, or severe/total), and accounting for the cluster sampling and the different selection probabilities.

survival times than other kinds of falls. They considered that mortality differences were more strongly associated with factors such as frailty and comorbidities, which predisposed the subjects to falling, than to the fall itself and/or any subsequent injuries. They concluded that inside falls could be a marker of frailty. Unfortunately, they did not compare survival with a control

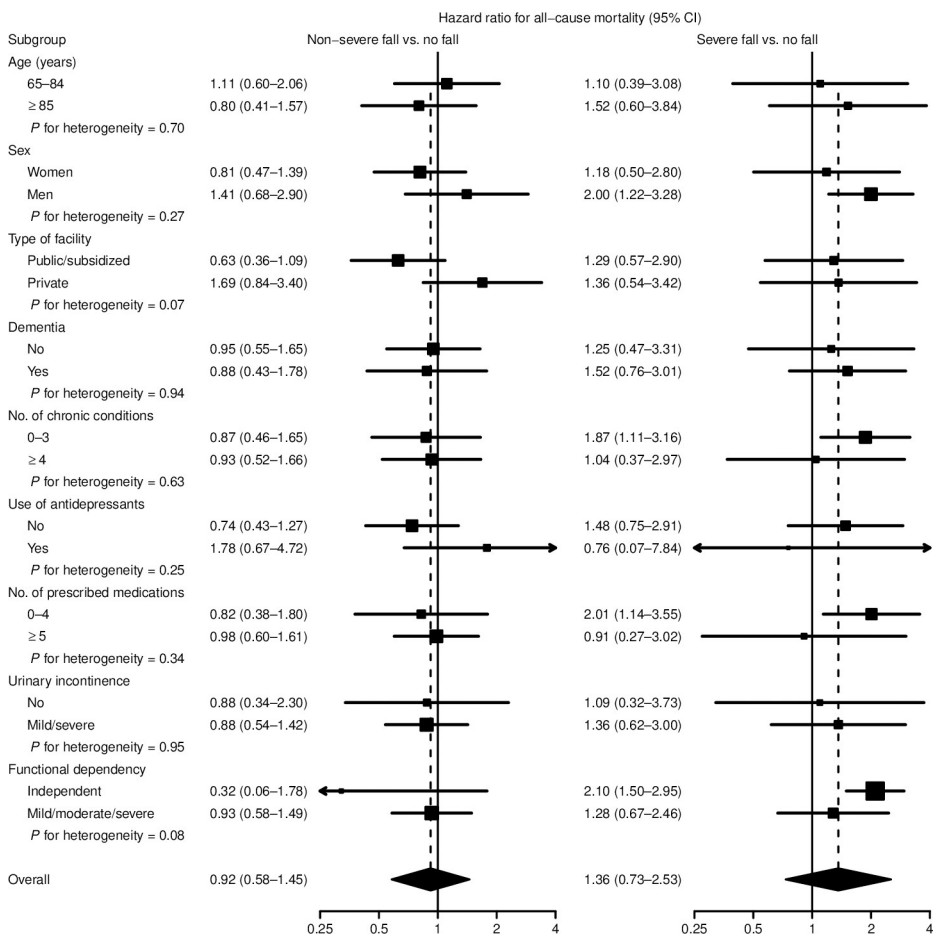

**Fig 3. Hazard ratios for all-cause mortality associated with severity of fall by subgroup of nursing home residents in Madrid, Spain.** Subgroup-specific hazard ratios (squares with area inversely proportional to the variance) and 95% confidence intervals (CIs, horizontal lines) were obtained from Cox proportional hazards models with interactions between severity of fall (no, non-severe, or severe) and the corresponding subgroup indicator, adjusted for baseline age (65–74, 75–79, 80–84, 85–89, or ≥ 90 years), sex (women or men), type of facility (public, subsidized, or private), dementia (yes or no), number of chronic conditions other than dementia (0–1, 2–3, or ≥ 4), use of antidepressants (yes or no), number of prescribed medications other than antidepressants (0–2, 3–4, or ≥ 5), urinary incontinence (no, mild, or severe) and functional dependency excluding (in)continence (independent, mild/moderate, or severe/total), and accounting for the stratified cluster sampling and the different selection probabilities.

group of non-fallers [19]. In line with this hypothesis, it is interesting to highlight that Wong *et al.* found that low falls ($\leq$0.5 meters) for older adults from hospital settings were an independent predictor of 1-year mortality when compared to higher-level falls. Moreover, for low fall injuries, they considered that the long-term cause of death was unlikely to be trauma related and regarded low falls mechanism as a surrogate marker of frailty. They concluded that frailty status could explain the increased mortality found by other studies [17]. We did not adjust for frailty but, besides age and comorbidities, we included functional dependency in our analysis which could be considered a close proxy for frailty. We did not find a relevant increase in mortality risk for residents who had suffered a fall. This held even when severity of the fall was accounted for (although there was slight non-significant increase in risk for severe falls). Thus, our results are consistent with Wong *et al.* conclusions [17]. Moreover, in our subgroup analyses, severe falls were associated with a marked increase in mortality risk in the case of residents with apparently better baseline states (fewer comorbidities, fewer prescribed medications, functionally independent) and men. A possible explanation is that in patients with worse baseline states, mortality is primarily associated with frailty rather than with the fall itself, which is consistent with other studies' conclusions [17, 19], and the impact of severe falls on mortality risk could be more marked on subpopulations with better baseline states. In line with this possible explanation, in those subpopulations with better baseline states, the reference group of non-fallers is expected to have good survival expectancies, so the mortality risk for those who suffered a severe fall could be remarkably higher when compared to them. On the contrary, in those populations with worse basal state (residents with frailty and more comorbidities) the reference group of non-fallers is supposed to have already bad survival expectancies. Previous studies have recommended the implementation of preventive measures aimed to attenuate frailty mainly in adults aged 75 years or older [8]. These could include, among others, the routine practice of moderate physical exercise [32] and a healthy eating pattern [33]. Our results are coherent with a possible benefit of primary preventive programs addressed to improve frailty in care homes residents. Further studies should assess the best preventive approaches of severe falls, for those in care homes residents with a better basal state.

In subgroup analyses, non-severe fallers, compared to non-fallers, presented a 69% increase in mortality risk in private facilities and a 37% decrease in risk in public institutions. However, we cannot provide a solid hypothesis for these differences in mortality risks. The important confounding factors have been controlled, so the reasons for a possible effect, if it really exists, are presumably due to differences in structural and organizational factors. Nevertheless, we do not have enough data to approach this analysis with the adequate rigor. Moreover, it is advisable to be cautious with the results from subgroup analyses since they are usually more affected by random and systematic errors. These results just suggest that future research should analyse in depth these possible associations.

Finally, in the specific case of hip fractures after an accidental fall in older populations, the hazard of death is greater during the subsequent 6–12 months and declines thereafter [34–39]. This is roughly consistent with our results which show a mild increase in risk of death for severe falls, mainly during the first year, and to a lesser extent during the second, with a progressive decrease thereafter.

A limitation of our study is that we only included residents who had suffered a fall during the month prior to the baseline interview in the analyses so we cannot rule out that some residents died shortly after the fall but before the interview (post-fall in-hospital mortality has been reported as between 3.7% and 16% [15, 16, 18, 40, 41]). Furthermore, residents who had suffered a fall and survived the first month could be stronger than non-fallers. Thus, our mortality rates after an accidental fall could be under-recorded, and the possible survival bias effect could hinder the identification of associations. However, as we were assessing long-term

survival, this possible survival effect by definition must be included in the outcomes in terms of survival. Likewise, patients who survived a fall and subsequently stayed in the hospital for over a month were not included, so our figures for residents who had suffered a fall in the month prior to the baseline interview could have been underestimated. Second, some deaths might not have been reported, something that would eventually generate non-differential mis-classification and, in general, lead to underestimation of the associations. Third, after a fall, mortality is higher in patients discharged into skilled nursing care or into a nursing home [15, 16]. As we studied care home residents, patients with better prognoses after a fall might not have been included in our study, but rather returned home instead of entering a care home. However, we adjusted for variables with a strong association with mortality after a fall, such as comorbid conditions, age, sex and functional dependency. Fourth, we cannot exclude that, in the last decade, the practice management regarding falls, could have changed in nursing homes leading to a possible improvement in the survival. Nonetheless, a possible recent improvement in the survival, would have been consistent with our results. Fifth, our main independent variable was to have suffered a fall within the month prior to the interview. Thus, we cannot ensure that, those classified as "non fallers", did not suffer a fall outside that period. In most of the studies from non-clinical settings with a long follow-up, the information of baseline falls was self-reported by asking the participant about having suffered a fall during the preceding 3 months or until the preceding year depending on the studies [20–22]. Donald *et al*. reported that they selected only the preceding three months, to self-report a previous fall, because they considered that the recall over a longer period was less accurate [20]. Anyhow, independently of the different timeframes for collecting the occurrence of a previous or subse-quent falls, our results were consistent with the rest of the literature that did not find signifi-cant associations between suffering only a single fall and long term mortality [20–22, 24]. Finally, it is worth mentioning that data on falls were gathered from the facility physicians (or nurses for 8% of residents), thus notably limiting the potential information bias.

The main strengths of the study are that we selected a representative cohort of the care home resident population, with a very high response-rate and including various types of facili-ties. These facts enhance our study's external validity. Moreover, the large sample size of the study limits random error. We have included the most important confounding variables in the analysis, such as urinary incontinence, comorbidities, age, sex and functional dependency [15, 40, 42–44]. Furthermore, these variables may collectively serve as proxies for unmeasured potential confounding factors, such as frailty.

## Conclusions

We found no association between having suffered a fall in the month prior to baseline inter-view and long-term overall survival in care home residents. Additionally, we did not find a marked association when taking into account the severity of the fall, although some associa-tions in selected subgroups warrant further attention.

## Author Contributions

**Conceptualization:** Alicia Padrón-Monedero, Roberto Pastor-Barriuso, Javier Damián.

**Formal analysis:** Alicia Padrón-Monedero, Roberto Pastor-Barriuso, Fernando J. García López, Pablo Martínez Martín, Javier Damián.

**Funding acquisition:** Javier Damián.

**Methodology:** Alicia Padrón-Monedero, Roberto Pastor-Barriuso, Javier Damián.

**Project administration:** Javier Damián.

**Writing – original draft:** Alicia Padrón-Monedero, Roberto Pastor-Barriuso, Fernando J. García López, Pablo Martínez Martín, Javier Damián.

**Writing – review & editing:** Alicia Padrón-Monedero, Roberto Pastor-Barriuso, Fernando J. García López, Pablo Martínez Martín, Javier Damián.

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
