## [Decision Letter · Decision Letter 0]

25 Feb 2020

PONE-D-19-20403

Falls and long-term survival among older adults residing in care homes

PLOS ONE

Dear Dr. Damián,

Thank you for submitting your manuscript to PLOS ONE. After careful consideration by 2 Reviewers and an Academic Editor, all of the critiques of both Reviewers must be addressed in detail in a revision to determine publication status. If you are prepared to undertake the work required, I would be pleased to reconsider my decision, but revision of the original submission without directly addressing the critiques of the two Reviewers does not guarantee acceptance for publication in PLOS ONE. If the authors do not feel that the queries can be addressed, please consider submitting to another publication medium. A revised submission will be sent out for re-review. The authors are urged to have the manuscript given a hard copyedit for syntax and grammar.

**Comments to the Author**

1. Is the manuscript technically sound, and do the data support the conclusions?

Reviewer #1: Yes

Reviewer #2: Yes

2. Has the statistical analysis been performed appropriately and rigorously? 

Reviewer #1: I Don't Know

Reviewer #2: I Don't Know

3. Have the authors made all data underlying the findings in their manuscript fully available?

Reviewer #1: Yes

Reviewer #2: Yes

4. Is the manuscript presented in an intelligible fashion and written in standard English?

Reviewer #1: Yes

Reviewer #2: Yes

5. Review Comments to the Author

Reviewer #1: This study investigated the association between having suffered a fall and long-term survival in nursing-home residents. The manuscript is interesting and well-written and the analysis are made based on a sufficiently large dataset. However, I have a few comments:

Introduction, lines 53 et seq.: It is written that falls may cause death. Please briefly describe which consequences may result from falls and lead to death.

Introduction, lines 61-62: Please name examples for minor and major injuries. This may relate to and be combined with my former comment.

Study population, lines 79-81: This study was based on data from 1998-1999 which is outdated. Please describe if practice regarding falls in nursing homes has changed since then and how it will impact findings and implications of the current study. It should be mentioned in the limitation section of the discussion, particularly since in the introduction it was argued that existing studies are not recent.

Study population, line 94 + line161: What is meant by “sampling weights” / “weighted by underlying population distribution”? Please explain how the weights were obtained and used.

Baseline data collection, lines 100-104: Unfortunately, falls of the nursing home residents were only recorded in the previous month of the interview at a random start in 1998/1999 which is a very short baseline. Therefore, only 12.6% of all residents had sustained a fall. A longer baseline would have enabled to choose nursing home residents because of their fall in order to increase the group of fallers. This design might be an important cause for the weak associations which should be mentioned in the limitation section of the discussion.

Baseline data collection, line 120: Why is urinary incontinence important to adjust for?

Baseline data collection: There is no information on whether any falls occurred during follow-up although it would affect the observed mortality. If the association of falls and mortality is to be investigated, why have additional falls not been observed? This information should either be added to the model or mentioned in the limitation section.

Statistical analysis: Please describe how the proportional hazards assumption for Cox proportional hazards models was tested and briefly report results.

Table 1: Please add the number and share of nursing home residents having sustained a fall in total and per fall severity group.

Discussion: There are several very long sentences which makes the otherwise interesting discussion difficult to read. Please split these in shorter sentences.

Reviewer #2: Considerations

1. Consider citing the questionnaire used in the interviews.

2. Consider citing a source or further explaining the rationale behind choosing 5 years as the cutoff for long-term mortality.

Questions

1. The authors report, "In subgroup analyses, non-severe falls were associated with a 69% increase in mortality risk 238 for residents in private facilities and a 37% decrease in risk for those in public institutions". Consider addressing this interesting and surprising finding in the discussion and adding any potential reasons for these results.

2. The authors report the increase in mortality within the first year following the interview and how this is in line with the literature. Was a hazard ratio calculated at the one-year mark as well as at 2-years? Would this have made a difference in the results?

3. Did/how did the authors ensure that those classified as "non fallers" did not suffer from a fall over the course of the study? If, for example, someone classified as a "non-faller" could have actually fallen two months before the interview or two months after the interview took place, how was this accounted for? I would highly encourage addressing this in the discussion.

6. PLOS authors have the option to publish the peer review history of their article (what does this mean?). If published, this will include your full peer review and any attached files.

**Do you want your identity to be public for this peer review?** For information about this choice, including consent withdrawal, please see our Privacy Policy.

Reviewer #1: No

Reviewer #2: No

We would appreciate receiving your revised manuscript by August, 2020. To enhance the reproducibility of your results, we recommend that if applicable you deposit your laboratory protocols in protocols.io, where a protocol can be assigned its own identifier (DOI) such that it can be cited independently in the future. For instructions see: http://journals.plos.org/plosone/s/submission-guidelines#loc-laboratory-protocols

We look forward to receiving your revised manuscript.

Kind regards,

Stephen D. Ginsberg, Ph.D.

Section Editor

PLOS ONE
---

## [Author Response · Author response to Decision Letter 0]

13 Mar 2020

Dear Editor,

We appreciate the suggestions and comments, which have helped us to improve the manuscript. Please find below the response to each individual comment and a detailed description of the changes made in the manuscript.

COMMENTS TO THE AUTHOR

1. Is the manuscript technically sound, and do the data support the conclusions?

Reviewer #1: Yes

Reviewer #2: Yes

2. Has the statistical analysis been performed appropriately and rigorously? 

Reviewer #1: I Don't Know

Reviewer #2: I Don't Know

3. Have the authors made all data underlying the findings in their manuscript fully available?

Reviewer #1: Yes

Reviewer #2: Yes

4. Is the manuscript presented in an intelligible fashion and written in standard English?

Reviewer #1: Yes

Reviewer #2: Yes

REVIEWER: 1 

COMMENTS TO THE AUTHOR.

We thank the reviewer for its suggestions, which give us the opportunity of improving the article.

 This study investigated the association between having suffered a fall and long-term survival in nursing-home residents. The manuscript is interesting and well-written and the analysis are made based on a sufficiently large dataset. However, I have a few comments:

Introduction, lines 53 et seq.: It is written that falls may cause death. Please briefly describe which consequences may result from falls and lead to death.

Introduction, lines 61-62: Please name examples for minor and major injuries. This may relate to and be combined with my former comment.

Authors: We agree that these two suggestions will improve the manuscript. In order to provide all this information we propose to include the following paragraph in the Introduction section (lines 60-67):

“Older adults have significantly higher mortality from minor injuries than those under 50s [12] and the interval from injury to death could be longer for minor or moderate injuries (for example rib fracture and chest or muscle contusions) than for more severe ones (hip, vertebral or skull fractures) [13]. Additionally, as the interval between injury and death increases, fall mortality could be underreported due to the death being attributed to a concomitant pathology [13, 14] (pneumonia, cerebral hemorrhage, pulmonary thromboembolism, worsening of comorbidities or other complication of the fall)”

Study population, lines 79-81: This study was based on data from 1998-1999 which is outdated. Please describe if practice regarding falls in nursing homes has changed since then and how it will impact findings and implications of the current study. It should be mentioned in the limitation section of the discussion, particularly since in the introduction it was argued that existing studies are not recent.

Authors: We have added the following paragraph to the Discussion section (strengths and limitations) (lines 368-371)

“Fourth, we cannot exclude that, in the last decade, the practice management regarding falls, could have changed in nursing homes leading to a possible improvement in the survival. Nonetheless, a possible recent improvement in the survival would have been consistent with our results. 

Study population, line 94 + line161: What is meant by “sampling weights” / “weighted by underlying population distribution”? Please explain how the weights were obtained and used.

Authors: At baseline, we oversampled residents in public/subsidized facilities and men. The weights were calculated in the usual way as the inverse of each participant’s selection probability and were intended to restore the representativeness of these groups according to the observed proportions in the targeted population.

To clarify the calculation of sampling weights, we have included the following sentence in the Materials and methods section (Study population, lines 97-101), supported by a new methodological reference (Levy PS, Lemeshow S. Sampling of Populations: Methods and Applications, third edition. New York: John Wiley & Sons, 1999):

“To correct for this oversampling, sampling weights were assigned to study participants as the inverse of their probabilities of selection (that is, the inverse of the sampling fractions by type of facility and sex) [25], so that the weighted distribution of these factors in the study sample matched their distribution in the entire population of care home residents.”

Please see also Materials and methods (Statistical analysis, lines 172-173) for the use of sampling weights in statistical analysis: 

“all analyses were weighted by the inverse of each participant’s selection probability to obtain unbiased estimates of population parameters…”.

Baseline data collection, lines 100-104: Unfortunately, falls of the nursing home residents were only recorded in the previous month of the interview at a random start in 1998/1999 which is a very short baseline. Therefore, only 12.6% of all residents had sustained a fall. A longer baseline would have enabled to choose nursing home residents because of their fall in order to increase the group of fallers. This design might be an important cause for the weak associations which should be mentioned in the limitation section of the discussion.

Authors: We agree with the reviewer that we cannot ensure that, those classified as "non fallers", did not suffer a fall outside the previous one month period. However, it has been described that, for the older people, suffering a fall is an intermittent experience that it is very likely to be completely resolved, rather than to evolve to regular falling one year later (Donald, I.P. and C.J. Bulpitt, The prognosis of falls in elderly people living at home. Age Ageing, 1999. 28(2): p. 121-5). Moreover, Donald et al. reported that they selected only the preceding three months to self-report a previous fall because they considered that the recall over a longer period was less accurate (Donald, I.P. and C.J. Bulpitt, The prognosis of falls in elderly people living at home. Age Ageing, 1999. 28(2): p. 121-5). Besides independently of the different timeframes for collecting the occurrence of a previous fall, our results were consistent with the rest of the literature that did not find significant associations between suffering only a single fall and long term mortality (Donald, I.P. and C.J. Bulpitt, The prognosis of falls in elderly people living at home. Age Ageing, 1999. 28(2): p. 121-5) (Dunn, J.E., et al., Mortality, disability, and falls in older persons: the role of underlying disease and disability. Am J Public Health, 1992. 82(3): p. 395-400) (Bath, P.A. and K. Morgan, Differential risk factor profiles for indoor and outdoor falls in older people living at home in Nottingham, UK. Eur J Epidemiol, 1999. 15(1): p. 65-73)( Sylliaas, H., et al., Does mortality of the aged increase with the number of falls? Results from a nine-year follow-up study. Eur J Epidemiol, 2009. 24(7): p. 351-5)

To explain these issues, we have now added the following comment to the Discussion section (lines 371-384).

“Fifth, our main independent variable was to have suffered a fall within the month prior to the interview. Thus, we cannot ensure that, those classified as "non fallers", did not suffer a fall outside that period. In most of the studies from non-clinical settings with a long follow-up, the information of baseline falls was self-reported by asking the participant about having suffered a fall during the preceding 3 months or until the preceding year depending on the studies [20-22]. Donald et al. reported that they selected only the preceding three months, to self-report a previous fall, because they considered that the recall over a longer period was less accurate [20]. Anyhow, independently of the different timeframes for collecting the occurrence of a previous or subsequent falls, our results were consistent with the rest of the literature that did not find significant associations between suffering only a single fall and long term mortality [20-22,24]. Finally, it is worth mentioning that data on falls were gathered from the facility physicians (or nurses for 8% of residents), thus notably limiting the potential information bias”.

Baseline data collection, line 120: Why is urinary incontinence important to adjust for?

Authors: Urinary incontinence is a potential confounder in this frail population because it is associated with falling and it is independently associated with mortality (Damian J, Pastor-Barriuso R, Valderrama-Gama E, de Pedro-Cuesta J. Factors associated with falls among older adults living in institutions. BMC Geriatr. 2013;13:6).

Please do see the Discussion section (lines 388-390).

“We have included the most important confounding variables in the analysis, such as urinary incontinence, comorbidities, age, sex and functional dependency. [15, 40, 42-44]”

Baseline data collection: There is no information on whether any falls occurred during follow-up although it would affect the observed mortality. If the association of falls and mortality is to be investigated, why have additional falls not been observed? This information should either be added to the model or mentioned in the limitation section.

Authors: We believe that the above paragraph, that has been included in the limitations section, also applies to this suggestion. 

Please, do see the Discussion section (lines 371-384).

“Fifth, our main independent variable was to have suffered a fall within the month prior to the interview. Thus, we cannot ensure that, those classified as "non fallers", did not suffer a fall outside that period. In most of the studies from non-clinical settings with a long follow-up, the information of baseline falls was self-reported by asking the participant about having suffered a fall during the preceding 3 months or until the preceding year depending on the studies [20-22]. Donald et al. reported that they selected only the preceding three months, to self-report a previous fall, because they considered that the recall over a longer period was less accurate [20]. Anyhow, independently of the different timeframes for collecting the occurrence of a previous or subsequent falls, our results were consistent with the rest of the literature that did not find significant associations between suffering only a single fall and long term mortality [20-22,24]. Finally, it is worth mentioning that data on falls were gathered from the facility physicians (or nurses for 8% of residents), thus notably limiting the potential information bias”.

Statistical analysis: Please describe how the proportional hazards assumption for Cox proportional hazards models was tested and briefly report results.

Authors: We tested for the proportional hazards assumption by using spline-based parametric survival models, which provide a similar but more efficient way to deal with non-proportional hazards than Cox semiparametric models (Royston, P. and M.K. Parmar, Flexible parametric proportional-hazards and proportional-odds models for censored survival data, with application to prognostic modelling and estimation of treatment effects. Stat Med, 2002. 21(15): p. 2175-97). Indeed, spline-based survival models allowed us to report not only the P values for the proportional hazards assumption, but also the estimated trends in hazards ratios over time. These results were already reported in Figure 2 and commented in the Results section of the original manuscript.

To further clarify this point, we have extended the description of the spline-based survival model and added a sentence on the test for the proportional hazards assumption in the Materials and methods section (Statistical analysis, lines 163-170):

“To allow for time-dependent effects of fall severity on mortality (that is, non-proportional hazards), smooth cumulative hazard ratios over time and 95% CIs were estimated from a spline-based parametric survival model [31], with the baseline log cumulative hazard as a natural cubic spline of log time with a single internal knot at the median failure time, interactions of spline terms with fall-severity categories, and adjustment for all baseline confounders. Proportional hazards were contrasted by using joint Wald tests for spline-by-severity interaction coefficients”.

In addition to the detailed results presented in Figure 2, we have also added the P values from the proportional hazards tests to the Results section (lines 232 and 235-236):

“(P for proportional hazards = 0.94)” and “…albeit this time-dependent pattern was not significant (P for proportional hazards = 0.70)”.

Table 1: Please add the number and share of nursing home residents having sustained a fall in total and per fall severity group.

Authors: We have included in Table 1 the numbers and percentages, of the overall population and by fall group, for the different baseline characteristics. If the reviewer wants us to provide the number of all fallers, no matter the severity, this information can be obtained by summing the non-severe and severe fall columns. To add a new column with this information we think that it could be confusing and somewhat redundant.”

Discussion: There are several very long sentences which makes the otherwise interesting discussion difficult to read. Please split these in shorter sentences.

Authors: Done.

REVIEWER: 2 

COMMENTS TO THE AUTHOR. 

We thank the reviewer for all its suggestions, which will give us the opportunity of improving the article.

Considerations:

1. Consider citing the questionnaire used in the interviews.

Authors: We acknowledge that citing the questionnaire used in the interviews could be of interest. Unfortunately, we cannot provide the cite of questionnaire because it is not published. 

2. Consider citing a source or further explaining the rationale behind choosing 5 years as the cutoff for long-term mortality.

Authors: Thank you very much for this suggestion. To detail this rationale for choosing the 5 years cut off, we have added the following paragraph to the Materials and methods section (lines 138-140):

“Our rationale for a 5-year follow-up was choosing a timeframe after which the occurrence of an event was hardly attributable to any change at baseline.”

Questions

1. The authors report, "In subgroup analyses, non-severe falls were associated with a 69% increase in mortality risk 238 for residents in private facilities and a 37% decrease in risk for those in public institutions". Consider addressing this interesting and surprising finding in the discussion and adding any potential reasons for these results.

Authors: We have included the following paragraph in the Discussion section (lines 334-343):

“In subgroup analyses, non-severe fallers, compared to non-fallers, presented a 69% increase in mortality risk in private facilities and a 37% decrease in risk in public institutions. However, we cannot provide a solid hypothesis for these differences in mortality risks. The important confounding factors have been controlled, so the reasons for a possible effect, if it really exists, are presumably due to differences in structural and organizational factors. Nevertheless, we do not have enough data to approach this analysis with the adequate rigor. Moreover, it is advisable to be cautious with the results from subgroup analyses since they are usually more affected by random and systematic errors. These results just suggest that future research should analyse in depth these possible associations”.

2. The authors report the increase in mortality within the first year following the interview and how this is in line with the literature. Was a hazard ratio calculated at the one-year mark as well as at 2-years? Would this have made a difference in the results?

Authors: We did calculate the hazards ratio at each of the 5 years of follow up. Please, do see Figure 2 showing the time dependent effects, with the hazard ratios (and 95% confidence intervals) at each of the 5 years of follow up.

3. Did/how did the authors ensure that those classified as "non fallers" did not suffer from a fall over the course of the study? If, for example, someone classified as a "non-faller" could have actually fallen two months before the interview or two months after the interview took place, how was this accounted for? I would highly encourage addressing this in the discussion.

Authors: Thank you very much for your comment, we acknowledge this as a limitation. 

We agree with the reviewer that we cannot ensure that, those classified as "non fallers", did not suffer a fall outside the previous one month period. However, it has been described that, for the older people, suffering a fall is an intermittent experience that it is very likely to be completely resolved, rather than to evolve to regular falling one year later (Donald, I.P. and C.J. Bulpitt, The prognosis of falls in elderly people living at home. Age Ageing, 1999. 28(2): p. 121-5). Moreover, Donald et al. reported that they selected only the preceding three months to self-report a previous fall because they considered that the recall over a longer period was less accurate (Donald, I.P. and C.J. Bulpitt, The prognosis of falls in elderly people living at home. Age Ageing, 1999. 28(2): p. 121-5). Besides independently of the different timeframes for collecting the occurrence of a previous fall, our results were consistent with the rest of the literature that did not find significant associations between suffering only a single fall and long term mortality (Donald, I.P. and C.J. Bulpitt, The prognosis of falls in elderly people living at home. Age Ageing, 1999. 28(2): p. 121-5) (Dunn, J.E., et al., Mortality, disability, and falls in older persons: the role of underlying disease and disability. Am J Public Health, 1992. 82(3): p. 395-400) (Bath, P.A. and K. Morgan, Differential risk factor profiles for indoor and outdoor falls in older people living at home in Nottingham, UK. Eur J Epidemiol, 1999. 15(1): p. 65-73)( Sylliaas, H., et al., Does mortality of the aged increase with the number of falls? Results from a nine-year follow-up study. Eur J Epidemiol, 2009. 24(7): p. 351-5)

Please, do see all this information detailed in the Discussion section (lines 371-384).

“Fifth, our main independent variable was to have suffered a fall within the month prior to the interview. Thus, we cannot ensure that, those classified as "non fallers", did not suffer a fall outside that period. In most of the studies from non-clinical settings with a long follow-up, the information of baseline falls was self-reported by asking the participant about having suffered a fall during the preceding 3 months or until the preceding year depending on the studies [20-22]. Donald et al. reported that they selected only the preceding three months, to self-report a previous fall, because they considered that the recall over a longer period was less accurate [20]. Anyhow, independently of the different timeframes for collecting the occurrence of a previous or subsequent falls, our results were consistent with the rest of the literature that did not find significant associations between suffering only a single fall and long term mortality [20-22,24]. Finally, it is worth mentioning that data on falls were gathered from the facility physicians (or nurses for 8% of residents), thus notably limiting the potential information bias”.

---

## [Decision Letter · Decision Letter 1]

30 Mar 2020

Falls and long-term survival among older adults residing in care homes

PONE-D-19-20403R1

Dear Dr. Damián,

We are pleased to inform you that your manuscript has been judged scientifically suitable for publication and will be formally accepted for publication once it complies with all outstanding technical requirements.

With kind regards,

Stephen D. Ginsberg, Ph.D.

Section Editor

PLOS ONE

Additional Editor Comments: Please consider following the suggestions of Reviewer #2 on updating citations.

**Comments to the Author**

1. If the authors have adequately addressed your comments raised in a previous round of review and you feel that this manuscript is now acceptable for publication, you may indicate that here to bypass the “Comments to the Author” section, enter your conflict of interest statement in the “Confidential to Editor” section, and submit your "Accept" recommendation.

Reviewer #1: All comments have been addressed

Reviewer #2: All comments have been addressed

2. Is the manuscript technically sound, and do the data support the conclusions?

Reviewer #1: Yes

Reviewer #2: Yes

3. Has the statistical analysis been performed appropriately and rigorously? 

Reviewer #1: Yes

Reviewer #2: I Don't Know

4. Have the authors made all data underlying the findings in their manuscript fully available?

Reviewer #1: Yes

Reviewer #2: Yes

5. Is the manuscript presented in an intelligible fashion and written in standard English?

Reviewer #1: Yes

Reviewer #2: Yes

6. Review Comments to the Author

Reviewer #1: (No Response)

Reviewer #2: I thank the authors for taking the time to address the comments.

Given that a history of a fall is one of the major risk factors of an injurious fall, it would further strengthen your paper to cite some more recent literature/nursing home-specific literature to support your statement that "our results were consistent with the rest of the literature that did not find significant associations between suffering only a single fall and long term mortality [20-22, 24]". Perhaps some relevant research in the area of long term care could be cited and more comparable to your population than the community population.

7. PLOS authors have the option to publish the peer review history of their article (what does this mean?). If published, this will include your full peer review and any attached files.

Reviewer #1: No

Reviewer #2: No

---

## [Editor Report · Acceptance letter]

28 Apr 2020

PONE-D-19-20403R1 

Falls and long-term survival among older adults residing in care homes 

Dear Dr. Damián:

I am pleased to inform you that your manuscript has been deemed suitable for publication in PLOS ONE. Congratulations! Your manuscript is now with our production department. 

With kind regards,

on behalf of

Dr. Stephen D. Ginsberg 

Section Editor

PLOS ONE